# Relative Age Effect in the Sport Environment. Role of Physical Fitness and Cognitive Function in Youth Soccer Players

**DOI:** 10.3390/ijerph16162837

**Published:** 2019-08-08

**Authors:** Florentino Huertas, Rafael Ballester, Honorato José Gines, Abdel Karim Hamidi, Consuelo Moratal, Juan Lupiáñez

**Affiliations:** 1Faculty of Physical Education and Sport Sciences, Catholic University of Valencia “San Vicente Mártir”, 46900 Torrent, Spain; 2Doctorate School, Catholic University of Valencia “San Vicente Mártir”, 46008 Valencia, Spain; 3Department of Experimental Psychology, and Mind, Brain and Behavior Research Center (CIMCYC), University of Granada, 18071 Granada, Spain

**Keywords:** RAE, youth sport talent selection, maturation, attention

## Abstract

The need to achieve short-term competitive outcomes in sports may influence the emergence of talent selection strategies, which could bias individuals’ opportunities. The present study aimed to further explore the relative age effect (RAE), a phenomenon that strongly influences youth sport development. The RAE refers to a disproportionately high percentage in sport teams of athletes born early in the selection year. Our primary focus was to explore whether the RAE is supported by behavioral evidence in favor of better fitness—and especially cognitive-attentional functioning—of early as compared to late-born players. A cross-sectional study was conducted on 105 young athletes (u10, *n* = 52; 9.8 ± 0.3 years old, and u12, *n* = 53; 11.8 ± 0.2 years old) attending two youth elite soccer academies. Attentional functioning, anthropometrics, physical fitness, and game intelligence were compared across two Age Groups (u10 vs. u12) and four Birth Quarters (BQ1–BQ4). The RAE was statistically significant (*p* < 0.001), showing that about 50% of participants were born in the first quarter and 75% were born in the first half of the year. More importantly, U12 players outperformed u10 players in measures that were related to sustained attention (with faster and less variable responses; *p* < 0.001 and *p* < 0.05, respectively), and in all anthropometric measures (*p* < 0.001), physical-fitness capacities (*p* < 0.05). Crucially, neither the attentional measures, game intelligence, anthropometrics, nor physical fitness were affected by BQ (all *ps* > 0.1 and BF_10_ between 0.08 and 0.6, showing strong evidence for the null hypothesis). The present findings suggest that the early selection process that occurs during scouting in youth soccer academies offsets the age-related differences that could be anticipated in cognitive skills, anthropometrics, and physical abilities, due to growth and maturation. These birth asymmetries could lead teams to disregard later maturation athletes and athletes born later in the year inducing a larger dropout of those players with the consequent reduction in the talent pool.

## 1. Introduction

After the abundant literature [1,2] that followed the description of “nature vs. nurture” by Francis Galton, further research is needed to an gain understanding of the relative contribution to human performance of inherited and environmental conditions. In sport, as in other contexts, the performance of athletes is the result of a complex interaction between genetic and environmental factors.

To excel in their sport disciplines, athletes experience a continuous adaptation process that impacts their physical and cognitive attributes. In fact, various studies have highlighted that the potential benefits of physical activity (PA) on cognitive functioning are modulated by the environmental characteristics of their sport context. Expert athletes have shown better performance than non-experts in sport-specific tests that assess memory, attention, information pick-up, anticipation, and decision-making skills [3].

The perceptual-cognitive adaptations that athletes may experience as a result of sport training might be constrained by their individual characteristics. In youth sport, one of the most influencing individual constraints is chronological age, while considering that athletes are mainly grouped by their date of birth in a selection year. This results in differences between the players of the same category of up to 12 months less one day in their age (i.e., relative age). The consequences of relative age in a group are known as the relative age effect (RAE). Studies from different fields, such as education [4] and sport [5], have shown that grouping learners by chronological age influences the way that learners/athletes adapt to the environment, providing advantages to some while harming others. A significant body of research aimed at evaluating birth date distribution in various sports has shown an overrepresentation of athletes that were born closer to the beginning of the selection year, that is, the first and second birth quarter (BQ), as compared to those born later (i.e., third and fourth BQ). Several studies have shown this asymmetry both in individual [6,7] and team sports [8,9,10]. Consistently, these asymmetries have been shown to be larger in youth teams as compared to elite teams [9,11], at a higher competition level (elite vs. non-elite) [12], and in sports where increased physical size and strength positively correlate with performance, such as ice hockey and soccer [13]. Interestingly, a few recent studies have shown that athletes born relatively late in the year score more and earn higher wages than their early-born peers [14,15,16], which has been called ‘RAE reversal’ [16].

Thus, the adaptation to a very demanding sport context, such as soccer, may imply divergent effects. One option is that differences between players born in the first and last quartiles of the year are accentuated because of the following: younger players are not competing in equal terms so their possibilities of being selected for the “best” teams are likely to be reduced, thus impacting their positive competitive experiences and their athletic self-concept [17]. This topic is not trivial, because previous findings indicate that relatively older soccer players 10–11 years of age were rated higher in ability and potential for success [18]. Another plausible option is that the sport environment exerts a homogenizing effect on individual differences, as younger athletes improve more to adapt to higher-competitive demands that are set by relatively older players. Focusing on athletes’ perceptual-cognitive performance, of primary interest in this study, and while attending to the underdog hypothesis [19], the following scenario is also possible: Younger or later development players must possess or develop superior technical/tactical or/and perceptual-cognitive skills to be selected and/or maintained in the team. This is a way of adapting to a competitive environment where at first they cannot take as much advantage as their peers of their physical attributes to solve different game situations.

Especially in soccer, there have been intense discussions about the RAE over the last decades. Yet, the vast majority of the studies have been descriptive attempts that are mostly focused on the presence or absence of this phenomenon [20]. The few studies aimed to assess the differences between athletes born at different times of the selection year focused mainly on variables that were related to the athletes’ physical attributes, such as anthropometrics and physical fitness abilities [21,22]. Their authors ignored the importance of cognitive characteristics that contribute to athletic success. Although anthropometric maturation seems to lead to the early selection process, there are interesting interrelations that need to be considered between body mass, fitness, and cognition. Many studies have highlighted the role of cardiovascular fitness through metabolic-physiological mechanisms, as the main mediator of PA associated with enhanced brain health and cognitive functions [23,24,25]. A recent review has discussed how physical activity and obesity, which are both variables that are associated with physical fitness, may work independently or together in affecting cognitive function [26]. A negative effect of overweight and fatness on general physical fitness has been described, specifically in youth soccer players [27,28].

However, a finding that is particularly interesting for the present study is that, in soccer and other sports, enhanced cognition and attention seem to be much more related to sport expertise than to fitness. The “cognitive component skills approach” understands sport training as a stimulating environment for experiencing brain plasticity and cognitive training in both domain-general and sport-specific cognitive skills. This rationale is in line with the “cognitive skill transfer” and the “broad transfer” hypotheses [29,30], which state that learning and practicing certain activities may lead to adaptations in basic cognitive abilities, which are potentially transferable to various skills in other domains. In this vein, recent findings have shown that externally-paced athletes (e.g., soccer players) showed better cognitive performance (i.e., vigilance and inhibitory control) than both non-athletes and self-paced athletes, without cardiovascular fitness modulation of these differential effects [31].

Thus, although several studies have highlighted the relevance of anthropometry and physical fitness for soccer performance [32], cognitive skills that are related to tactical and game intelligence are becoming more relevant in the scouting of young talents in soccer [33,34]. Therefore, this complex pattern of interrelations between growing, physical fitness, and cognitive functioning should also be considered in the early selection of athletes, along with anthropometric and fitness measures of early development and maturation.

While considering all of this, the main purpose of our research was to explore the relationship between the RAE and attentional functioning in two elite soccer academies for the u10 and u12 age groups, taking a step forward from the vast majority or studies that have approached this topic exclusively from the anthropometric and physical fitness perspectives. According to previous literature, we could expect to find birth asymmetries that are motivated by a highly competitive environment when short-medium performance was prioritized over the long-term potential of relatively younger or less mature players. More importantly, and in line with previous findings [35] that have shown that maturity bias emerges from 11 years onwards, we expected to find higher birth asymmetries in the u12 teams mainly in regards to better physical performance. However, according to the underdog hypothesis, we expected to find differences in cognitive performance between the u10 and u12 groups but not within each age group.

## 2. Materials and Methods

### 2.1. Study Design and Participants

A cross-sectional study was conducted. The sample included one hundred and five (105) young male soccer players aged 9.2–12.2 years (10.8 ± 1.0 years old) that were enrolled in two youth elite academies of LaLiga clubs in the Valencia region of Spain. The best two u10 teams (*n* = 52; 9.8 ± 0.3 years old) and two u12 teams (*n* = 53; 11.8 ± 0.2 years old) in each club participated in the study. The cut-off date that was fixed by Fédération Internationale de Football Association (FIFA) and the Spanish Football Federation from January 1st to December 31st was followed. Nevertheless, the analyses of attentional performance were performed on data from 87 participants, after excluding participants who could not perform the task appropriately. A sensitivity analysis was conducted while using G* power [36]. It showed that with our sample size (*n* = 87) divided into eight groups (i.e., four birth-date quarters in each Age Group), the minimum effect size that could be detected for α = 0.05 and 1 − β = 0.80 was f = 0.30 (i.e., minimum detectable effect).

Participation in the study was voluntary and all of the participants and their parents or legal guardians were properly informed regarding the risks and benefits of the study prior to any data collection and signed an institutionally approved informed consent document. All of the participants were informed about their right to leave the experiment at any time. The managers of the clubs were debriefed in regards to the purposes of the study and given an explanation of the main results with easily understandable data. The experiment was conducted in accordance with the ethical standards of the 1964 Declaration of Helsinki (last update: Seoul, 2008) as part of a larger research project that was approved by the University of Granada Ethics Committee (175/CEIH/2017).

### 2.2. Apparatus, Materials and Procedure

The participants were evaluated in three different sessions (Attentional Assessment Session, Anthropometrical Assessment, and Physical Fitness Assessment Session) in a counterbalanced order. Testing was performed in the first training session of the microcycle, scheduled at least 48 h later than the previous training session or competition match. All the data were collected under standard conditions of time (from 4 pm to 7 pm) and temperature (ranging from 13 °C to 23 °C) from February to May 2018. The participants were cited 60 min. before the training session and were assessed in the media room (Attentional Assessment Session), medical room (Anthropometrical Assessment) and soccer pitches (Physical Fitness Assessment Session) of their club’s training facilities.

#### 2.2.1. Characteristics of Players

The baseline anthropometrical variables of players included height and weight. They were measured and tested at the beginning of the first testing session. Participants’ chronological age (CA) was the difference between their date of birth and the date of the Attentional Assessment Session. Data from each team were collected within two-week periods. To test the RAE, the chronological age birth dates of players were divided into Birth Quarters (BQ): the first quarter included January, February, and March (BQ1), the second quarter included April, May, and June (BQ2), the third quarter included July, August, and September (BQ3), and the fourth quarter included October, November, and December (BQ4).

#### 2.2.2. Attentional Assessment

The Attentional Networks Test for Interactions and Vigilance—executive and arousal components (ANTI-Vea) developed by Luna and colleagues [37] was used to collect data regarding attentional functioning. The ANTI-Vea is a behavioral task that was developed to analyze the classic attentional functions that were measured by the ANTI task [38] (i.e., phasic alertness, orienting and executive control and their interactions), with the additional assessment of executive vigilance (EV) and arousal vigilance (AV). The ANTI-Vea task included three different types of trials: ANTI (60%), EV (20%), and AV (20%).

Participants sat on a chair 60 cm from a computer monitor in a dimly lit and noise-reduced room and completed the ANTI-Vea task at the same time. The online version of the task was used (https://www.ugr.es/%7Eneurocog/Sitio_web/ANTI/). Participants completed several practice blocks with visual feedback and three blocks of experimental trials with standard levels of noise (2), difficulty (2), and stimuli duration (400 ms), and the responses were collected while using various PC laptops (15-inch color screen HP Pro Book). The participants were encouraged to keep their eyes on the fixation point all the time.

The following stimuli were used in the ANTI and EV trials: a black fixation cross (~7 pixels, px), a black asterisk (~13 px), a warning tone (2000 Hz), and five black arrows (50 px wide × 23 px high each), pointing either left or right. Each arrow was horizontally separated by ~63 px from the adjacent arrows. In each trial, a random variability of ±2 px was applied to the horizontal and the vertical position of each arrow to make it more difficult to find the infrequent displaced targets. In the EV trials, the vertical displacement of the central arrow was fixed at 8 px.

In the ANTI trials (60% of the trials), the participants had to respond according to the direction of the central arrow target (“C” for left and “M” for right), while ignoring the flanking arrows. The interference variable was defined according to the congruency of the direction of the flanking and target arrows: congruent trials (50% of trials), when the target was flanked by arrows pointing in the same direction, and incongruent trials (the other 50% of trials), when the flanking arrows and the target pointed in opposite directions. The orienting signal was presented in two-thirds of the trials above or below the fixation point. Three orienting conditions were established according to the presence of the cue: cued location trials, when the cue was presented at the same location as the target; uncued location trials, when the cue was presented at the opposite location to the target; and, No-cue trials, when the cue was not presented. The alerting signal was presented before the onset of the target in only half of the trials. The alerting variable was established according to the presence (i.e., tone) or absence (i.e., no tone) of the alerting sound (See Figure 1a). For the ANTI trials, the mean and SD of RT, error percentage, and attentional indexes of alertness (No Tone—Tone trials, exclusively in No-cue trials), orienting (Uncued—Cued trials), and executive control (Incongruent—Congruent trials),as defined by Callejas and colleagues [38], were obtained for both RT and error percentages.

In the EV trials (20% of the trials), the same procedure was followed, except that the target was vertically displaced from the central position. The participants had to detect the vertical displacement by pressing the spacebar, while ignoring the direction of the target (See Figure 1b). Data from the various conditions of the warning signal, visual cue, and congruency variables were collapsed and not taken into account in the analyses. HITs were calculated as the percentage of displaced targets that were identified correctly. False alarms (FAs) were defined as the proportion of space bar responses (i.e., the response for infrequent stimuli) given to non-displaced targets. Finally, the mean RT and SD of RT were calculated.

In AV trials (20% of the trials), no tone, visual cues, or arrows were presented. These trials started in the same way as the ANTI and EV trials, and then the fixation point remained fixed in the screen for 500 ms (i.e., the same duration as the tone plus the visual cue signals in the ANTI and EV trials). Next, a red millisecond counter appeared in the center of the screen, starting at 1000 and going down to zero. The participants were asked to stop the counter as fast as they could by pressing any key of the keyboard (see Figure 1c). The mean and SD of RT were considered.

Before the experimental task, participants completed several practice blocks with visual feedback. First, they were given instructions to perform the ANTI trials, with a practice block of 16 such trials. Next, they received instructions to complete the EV trials, with a practice block of 32 randomized trials (16 ANTI and 16 EV trials). After that, they received instructions to complete the AV trials, followed by a practice block of 48 randomized trials (16 ANTI, 16 EV, and 16 AV trials). Finally, the participants performed a last practice block of 40 randomized trials (24 ANTI, 8 EV, and 8 AV trials) without visual feedback. At this point, if the participants still had any doubts, they could ask questions or perform the last practice block again. Otherwise, they continued with the experimental section of the task, which included three blocks of 80 randomized trials each (48 ANTI, 16 EV, and 16 AV trials per block), with no pause and no visual feedback.

#### 2.2.3. Physical Fitness Assessment

All of the physical fitness tests were performed between 5 and 8 pm during the first training session of the week (at least 48 h later than the previous training session or competition match). Ambient temperature was 17.3 ± 3.2 °C and humidity was 75 ± 4%. Prior to the testing, the participants followed the same supervised warm-up procedure with 10 min. of jogging, change of direction, and sprint drills. Finally, the participants performed a modified version of the agility T-test, a 24-m speed test, and the Leger Multi-stage fitness test.

##### Agility: Modified T-test

The T-test is a useful agility test for the assessment of multidirectional movement (i.e., forward, lateral, and backward) and leg power, leg speed, and agility [39]. A marked course on the soccer field using four cones was set up, as depicted in Figure 2. A photocell gate system (SportsMetrics, Valencia, Spain) that was connected to an electronic chronograph with 1 ms accuracy (i.e., 1000 Hz sampling frequency) (SportsMetrics, Valencia, Spain) was used to record the time. The infrared beam of each photocell was placed 0.9 to 1.0 m from the ground, depending on the player’s height, so that the photocell beam cut at the hip level.

Following previous recommendations [40], the athletes completed a modified agility T-test. At the starting position, the player placed his foot just behind the start line, with the knee of the advanced limb in semi-flexion and the arms flexed to avoid cutting the infrared beam. The forward body movement cutting the infrared beam started the stopwatch. After explaining the purpose of the T-test, the experimenter described and demonstrated the proper route and technique. Athletes had to keep their body facing forward at all times and physically touch each cone with the correct hand. On the experimenter’s command, the participants sprinted forward from the start at cone A to cone B and touched the top of the cone with their right hand. Subsequently, they shuffled left to cone C and touched it with their left hand. After that, they shuffled right to cone D and touched it with their right hand. They shuffled back to cone B and touched it with their right hand before running backward to pass the start gate (cone A). All of the participants underwent three practice trials before performing the test. The goal of the test was to complete the course as quickly as possible. Two maximal trials were completed with three minutes of recovery between them. The time of both trials was recorded, but only the fastest time was considered for later statistical analyses.

##### Speed: 24-m Speed Test

The speed test consisted of a 24-m track that participants had to run through as fast as possible. The running times were measured while using the same timing gate photocell system used in the previous test (SportsMetrics, Valencia, Spain). In this test, two photocell gates were used—one at the starting line and one at the finish line. The participants completed one practice trial and two maximal testing trials with three minutes of recovery between them. The best time was used for the statistical analyses.

##### Endurance: Leger Multi-Stage Fitness Test

The Leger Multi-stage fitness test was originally designed to determine the maximal aerobic power of school children and healthy adults [41]. This test and its adaptations are one of the most common assessments for measuring cardiorespiratory fitness in studies that involved young participants [42]. All of the athletes of the team ran back and forth on a 20 m course and had to reach the 20 m line at an initial speed of 8.5 km/h that progressively increased (0.5 km/h) according to a pace that was dictated by a pre-recorded tape. The test finished when the participants acknowledged voluntary exhaustion or were not able to follow the pace during two consecutive acoustic signals. The time completed was used to define the time-to-exhaustion (TTE).

#### 2.2.4. Game Intelligence

Members of the staff in each team, who were accredited by the Spanish Football Federation, evaluated the soccer game intelligence of each player in the team while using a five-point scale: 1 = very weak, 2 = weak, 3 = average, 4 = good, and 5 = very good. Scores were given according to the player´s ability to make a fast decision adapted to the game context. Coaches were informed regarding the relevance of using the full range of scores among the players of the same team, without comparing them with other teams. The mean values of all the data that were obtained from all the coaches were analyzed.

### 2.3. Statistical Procedure

Descriptive statistics (mean ± standard deviation, SD) were used to provide information regarding various aspects of the sample and groups. Independent variables included Age Group (u10 vs. u12) and Birth Quarter (BQ1, BQ2, BQ3 and BQ4). The RAE based on players’ distribution by BQ was analyzed while using chi-square tests. Age Group and BQ effects on the different dependent variables were analyzed while using univariate ANOVAs. When the variables violated the assumptions of normality and sphericity, non-parametric tests (i.e., Kruskall Wallis) were used.

A first analysis was conducted on the ANTI trials by means of repeated measures ANOVAs on both mean RT and percentage of errors, including Alerting signal (No tone/Tone), Orienting visual cue (Invalid/No Cue/Valid), and Congruency (Congruent/Incongruent). Next, an index was computed for each attentional network and dependent variable (RT and error percentage).

For the EV trials, the percentages of hits and false alarms (FA) were computed. Means and SD of RT were also obtained. For the AV trials, means and SD of RT were also calculated.

Further Bayesian analyses were performed on non-significant interactions. We tested the main effect model against a null model reporting Bayes factors (BF_10_), which quantify the power to evaluate the evidence in the data supporting the absence of a main effect or interaction.

The alpha level was set at *p* < 0.05 for univariate ANOVAs, repeated measures ANOVAs, and chi-square tests. The partial eta squared (*η_p_^2^*) effect size is reported, which indicated small (*η_p_^2^* > 0.01), moderate (*η_p_^2^* > 0.06), or strong (*η_p_^2^* > 0.14) [43] effects. Bayesian’s results were interpreted according to the discrete categories of evidential strength that was proposed by previous literature [44]. Statistical procedures were carried out in JASP computer software (Version 0.9, JASP Team, Amsterdam, The Netherlands).

## 3. Results

### 3.1. Preliminary Analysis

Analyses of chronological age, RAE, anthropometrics, and physical fitness abilities included data from all of the participants who fulfilled the requirements in each assessment session. Nevertheless, for the analyses of cognitive functioning, data from participants with more than 35% of errors in the ANTI trials were excluded. For RT analyses, trials with incorrect responses (8.19%) and those with an RT shorter than 200 ms (1.85%) and longer than than 1500 ms (1.78%) were excluded. Finally, analyses of attentional performance were performed on data from 87 participants. Table 1 displays descriptive statistics of participants’ characteristics according to Age Group and BQ.

### 3.2. Attentional Functioning

A first analysis was carried out on mean RT and error percentages in ANTI trials to test the general functioning of the ANTI-Vea task in this population regarding the measurement of the three attentional networks. Separate 2 (Alerting) × 3 (Orienting) × 2 (Congruency) repeated measures ANOVAs were performed on mean RT and error percentages. RT analyses showed the typical pattern of results, that is, the main effects for each variable, *F*(1, 86) = 39.821, *p* <0.001, *η_p_^2^* = 0.316, *F*(2, 172) = 45.783, *p* < 0.001, *η_p_^2^* = 0.347, and *F*(1, 86) = 197.929, *p* < 0.001, *η_p_^2^* = 0.697, respectively, for Alerting, Orienting, and Congruency, respectively, and the usual Alerting x Congruency, *F*(1, 86) = 16.774, *p* < 0.001, *η_p_^2^* = 0.163 (larger flanker-congruency effect with the alerting signal) and Orienting × Congruency interactions, *F*(2, 172) = 10.299, *p* < 0.001, *η_p_^2^* = 0.107 (smaller flanker-congruency effect in cued than uncued trials). The Alerting x Orienting interaction did not reach statistical significance, *p* = 0.241). The corresponding analysis performed on the error percentages showed similar results with *F*(1, 86) = 5.516, *p* = 0.021, *η_p_^2^* = 0.061, *F*(2, 172) = 6.201, *p* = 0.003, *η_p_^2^* = 0.068, and *F*(1, 86) = 68.355, *p* < 0.001, *η_p_^2^* = 0.446 for the main effects of Alerting, Orienting, and Congruency, respectively. Similarly, the usual Orienting x Congruency interaction was also significant, *F*(2, 172) = 9.833, *p* < 0.014, *p* < 0.001, *η_p_^2^* = 0.104. The Alerting x Orienting and Alerting x Congruency interactions did not reach statistical significance (*p* = 0.148, and *p* = 0.114, respectively). Two indexes were computed for each attentional network—one for RT and one for error percentage—to further analyze the alertness, orienting, and cognitive control as a function of Age Group and BQ. All the indexes and the effect of these variables on each measure are presented in Table 1. Alertness was computed as No Tone—Tone exclusively in the No Cue condition (following Callejas et al. 2004), Orienting was computed as Invalid—Valid, and Control was computed as Incongruent—Congruent.

### 3.3. Age Group and BQ

The distribution of players according to BQ significantly differed, χ^2^ (3, *n* = 105) = 36.45, *p* < 0.001, showing that players that were born in BQ1 were overrepresented (49.5%), followed by players born in BQ2 (22.9%) and BQ3 (16.2%). Only 11.4% of the players were born in the fourth BQ4. The RAE effect was present in both age categories u10, χ^2^ (3, *n* = 52) = 14.92, *p* = 0.002, and u12, χ^2^ (3, *n* = 53) = 22.85, *p* < 0.001 (see Table 1 and Figure 3). The distribution of players by BQ was the same when analyses was performed only with the 87 players who had completed the cognitive assessment, χ^2^ (3, *n* = 87) = 38.10, *p* < 0.001, showing that players born in BQ1 were overrepresented (52.9%), followed by players born in BQ2 (21.8%), BQ3 (13.8%), and BQ4 (11.5%). The RAE effect was similar in both age categories u10, χ^2^ (3, *n* = 42) = 17.05, *p* < 0.001, and u12, χ^2^ (3, *n* = 45) = 21.58, *p* < 0.001.

#### 3.3.1. Age and Anthropometrics

The results showed the expected main effects of Age Group and BQ on Chronological Age, *F*(1, 97) = 8667.3, *p* < 0.001, *η_p_^2^* = 0.989, and *F*(3, 97) = 264.20, *p* < 0.001, *η_p_^2^* = 0.891, respectively. They indicated that, obviously, u10 players were on average 2.03 years younger than the u12 players, and that players that were born in BQ1 were younger that those that were born in BQ2, *t* = 9.95, *p* < 0.001); the same pattern was found when comparing BQ2–BQ3 and BQ3–BQ4, *t* = 9.31, *p* < 0.001, and *t* = 6.56, *p* < 0.001, respectively.

Height, *F*(1, 92) = 56.705, *p* < 0.001, *η_p_^2^* = 0.381, and Weight, *F*(1, 92) = 29.019, *p* < 0.001, *η_p_^2^* = 0.240, were significantly modulated by the Age Group, showing that the u12 players were 12.5 cm taller and 7.86 kg heavier than players of u10 teams. Interestingly, neither of these anthropometrical variables were modulated by BQ, *p* = 0.174 for Height and *p* = 0.578 for Weight.

However, the lack of evidence or significance in frequentist analyses cannot be taken as evidence supporting the lack of an effect. Bayesian analysis are especially relevant in this case. They reveal whether data provide evidence favoring either the alternative hypothesis (with a larger Bayes Factor—BF_10_—providing stronger evidence) or the null hypothesis (with a lower BF_10_ supporting it more strongly), or no evidence (BF_10_ between 0.33 and 3) [45]. Importantly, as can be observed in Table 1, the Bayesian ANOVA showed anecdotal (Height BF_10_ = 0.65) and moderate (Weight, BF_10_ = 0.133) evidence for the null hypothesis (H_0_; no effect of Birth Quarter), which suggested that BQ did not have any effect on these variables (H_1_).

#### 3.3.2. Physical Fitness

Our results showed the predictable main effects of Age Group on all the physical fitness abilities. U10 players were 0.95’’ slower than u12 ones in the Agility test, *F*(1, 93) = 16.01, *p* < 0.001, *η_p_^2^* = 0.147, and 0.351’’ slower in the Speed test, *F*(1, 93) = 49.24, *p* < 0.001, *η_p_^2^* = 0.346. In the same vein, u12 players sustained their effort 2.49’ longer than their u10 counterparts in the Endurance test, *F*(1, 78) = 38.54, *p* < 0.001, *η_p_^2^* = 0.331. Yet, importantly, none of the physical fitness variables were affected by BQ: Agility, *p* = 0.528, Speed, *p* = 0.678, and Endurance, *p* = 0.537). In addition, the Bayes Factor ANOVAs again revealed moderate evidence for the null hypothesis that Birth Quarter had no effect on physical fitness abilities (Agility, BF_10_ = 0.157; Speed, BF_10_ = 0.114; and, Endurance, BF_10_ = 0.187).

#### 3.3.3. Game Intelligence

No Age Group effect, *p* = 0.980, or BQ effect, *p* = 0.313 was found in players’ game intelligence. The Bayesian ANOVA showed anecdotal evidence for the null hypothesis (H_0_; no effect of Age Group, BF_10_ = 0.206, or Birth Quarter, BF_10_ = 0.202, on game intelligence).

#### 3.3.4. Cognitive Functioning

Regarding attentional functioning that was measured by the ANTI trials, our results only revealed a main effect of Age Group on mean RT, *F*(1, 85) = 14.19, *p* < 0.001, *η_p_^2^* = 0.143, showing that u12 players were 107 ms faster than u10 players. The mean SD RT and Control Index Error rate were marginally moderated by the Age Group, *p* < 0.071 and *p* < 0.071, respectively. None of the other ANTI variables were modulated by the Age Group (all *p*s > 0.178) or BQ (all *p*s > 0.209). Regarding the Age Group effect, Bayes factor ANOVAs suggested anecdotal evidence (ANTI Mean RT SD, BF_10_ = 0.5; ANTI Mean Error Rate, BF_10_ = 0.452, Orienting Index RT, BF_10_ = 0.501, Control Index RT, BF_10_ = 0.372, and Control Index Error Rate, BF_10_ = 0.921) or moderate evidence (Alertness Index RT, BF_10_ = 0.226; Alertness Index Error Rate, BF_10_ = 0.27; and, Orienting Index Error Rate, BF_10_ = 0.232) for the null effect of Age Group on cognitive functioning. Similarly, Bayesian ANOVAs suggested strong (ANTI Mean RT, BF_10_ = 0.086; ANTI Mean RT SD, BF_10_ = 0.097; and, Control Index RT, BF_10_ = 0.082), moderate (ANTI Mean Error Rate, BF_10_ = 0.146; Alertness Index RT, BF_10_ = 0.282; Alertness Index Error Rate, BF_10_ = 0.185; Orienting Index RT, BF_10_ = 0.102; and, Orienting Index Error Rate, BF_10_ = 0.102), and anecdotal (Control Index Error Rate, BF_10_ = 0.377) evidence for the null effect of Birth Quarter on cognitive functioning.

Regarding Executive Vigilance, as measured in the EV trials, our analyses found that the Age Group had a significant effect on Mean RT and Mean RT SD in the HIT trials, *F*(1, 85) = 10.73, *p* = 0.002, *η_p_^2^* = 0.112 and *F*(1, 85) = 5.404, *p* = 0.022, *η_p_^2^* = 0.060, showing that u12 players were 115 ms faster and 17 ms more stable than u10 players. Hits (*p* = 0.695) and FA (*p* = 0.293) were not influenced by Age Group. By contrast, the Birth Quarter only modulated Mean RT SD HIT, *F*(3, 83) = 2.821, *p* = 0.044, *η_p_^2^* = 0.093. Post-hoc tests showed a smaller response variability (25 ms) in players that were born in BQ2 than in those born in BQ1, *t* = 2.775, *p* < 0.032. None of the other EV variables analyzed were influenced by BQ (all *p*s> 0.518). Bayesian ANOVAs provided moderate (Hits, BF_10_ = 0.24) and anecdotal (False Alarms, BF_10_ = 0.366) evidence for the null effect of Age Group on the functioning of Executive Vigilance. Similarly, Bayesian analyses suggested strong (Hits, BF_10_ = 0.092; and False Alarms, BF_10_ = 0.085) and moderate (Mean RT HIT, BF_10_ = 0.172) evidence of no effect of Birth Quarter on VE.

Finally, regarding Arousal Vigilance, our analyses revealed a main effect of Age Group on both AV variables: AV RT, *F*(1, 85) = 22.22, *p* < 0.001, *η_p_^2^* = 0.207, and AV SD, *F*(1, 85) = 22.22, *p* < 0.001, *η_p_^2^* = 0.153. U10 players had slower (93 ms) and more variable (34 ms) responses than u12 players in AV trials. Birth Quarter did not modulate any of the AV variables, mean RT (*p* = 0.556) or mean SD (*p* = 0.473). Bayesian ANOVAs provided moderate evidence (AV RT, BF_10_ = 0.172; and AV SD, BF_10_ = 0.204) for the null effect of Birth Quarter on Arousal Vigilance. 

Figure 4 shows the mean standardized Z-scores for anthropometrics, physical fitness, and cognitive measures, which suggested a statistically significant main effect of Age Group. Bivariate correlations between all relevant variables in our study are presented in Appendix A. Although the analysis of the specific relations between all the studied variables is beyond the scope of the present study, the correlation analysis is included in the Appendix A for the sake of completeness. In general, the effects of Age and Birth Quarter showed a similar pattern of data as that reported above. Specifically, strong correlations were observed between chronological age and anthropometrics, physical fitness, and sustained attention (arousal vigilance) variables in the overall sample. However, this pattern completely disappeared within each age group. Futhermore, interesting relations emerged between different variables within each domain (anthropometrics, physical fitness, and attentional functioning variables), which deserves further exploration in future studies.

## 4. Discussion

Our study is the first attempt to explore the relationship between RAE, fitness and cognitive performance in youth soccer academies, going a step forward from the vast majority of studies that explore this topic. The main purpose of our research was to explore the relationship between the RAE and physical and cognitive performance in soccer players from two elite academies of u10 and u12 age groups. In the sport environment, in general, but specifically in soccer, the RAE has been attributed to the influence of bio- physio- psychological maturity status and its effect on different variables that are associated to sport-specific performance (i.e., height, weight, strength, endurance, and power) [46,47]. The novelty of our approach to the study of the RAE is that, far from merely describing the presence of this effect, it addresses not only its relationship with traditionally evaluated physical variables, but also other behavioral variables, such as attentional functioning and game intelligence, which have not been previously explored in relation to the RAE.

Our sample of participants was asymmetrically distributed regarding BQ, showing robust birth asymmetries. About 50% of the players in each team had been born during the first quarter of the year, and close to 75% of participants had been born in the first half of the year. These data agree with numerous previous findings that have shown an overrepresentation of athletes that were born at the beginning of the selection year when compared to those born at the end of the year [12,13,47,48,49,50].

Regarding the effect of age (or Age Group), our results confirm our hypothesis regarding the u12 players outperforming the u10 players in all the anthropometric (i.e., weight and height) and physical-fitness capacities (i.e., agility, speed, and aerobic endurance). These findings agree with those of previous research [15,16]. Importantly, however, and consistent with previous studies, neither the general anthropometrics and physical fitness variables nor soccer-specific game intelligence rated by the coaches [50] were affected by BQ, with the Bayesian analyses strongly supporting the absence of some of these effects. 

A recent follow-up study [51] suggested that advanced maturity status in players that were born in the second half of the selection year could explain the absence of BQ effects on the variables studied. The youngest players selected, born in the second half of the year, may compensate/counteract the chronological age effect by being early maturators, that is, by matching their levels of physical development to those that were born in the first mid-year [48,52]. Therefore, inter-individual variation in biological maturity status is one of the major confounders among athletes when they are classified on the basis of chronological age criteria [51]. However, while players that were born in the last half of the year are less represented in most sport contexts, it has been described that their earlier maturation could facilitate their development and future successful sport career [51,52]. It seems that, at least during the first stages of talent selection, soccer systematically excludes late maturing players, favoring early maturing ones as chronological age and sport specialization increase.

Regarding the functioning of the attentional networks, it should be noted that the online ANTI-Vea task worked well in our sample of young soccer players, replicating the main effects (alertness, orienting, and executive control) and interactions (orienting x executive control, alertness x orienting, and alertness x executive control) reported by previous research while using the ANT [53], ANT-I [38,54] and ANTI-Vea [37]. However, no effects of Age or BQ on the functioning of the attentional networks were found. This finding could be explained by previous research regarding the maturation of the attentional networks, showing that its peak development is reached around the age of nine years [54,55,56,57]. Participants in our study ranged from 9.2 to 12.2 years, and are thus likely to have reached or passed their peak maturation period regarding the development of the attentional networks.

Behavioral differences between the groups were only found in several measures that were related to sustained attention (mean RT and SD RT), with u12 players generally showing faster and less variable responses than u10 players. This pattern of results has been previously described [58], showing that differential developmental changes at these ages induce a reduction in mean reaction time and response variability. 

Previous studies have shown superior vigilance capacities (mean RT and RT SD) in children [59], adolescent soccer players [60], and young athletes in various externally-paced sports [31,61] when compared to non-athletes and athletes that are involved in self-paced sports. In the same vein, other studies have shown greater cognitive flexibility [62] and motor inhibition and a larger alerting effect in the ANT [63,64] in elite as compared to amateur soccer players aged 8–17 years. From these findings, one could expect better motor inhibition and alerting in relatively older soccer players as a result of greater development and more soccer experiences than in younger players. However, it is important to note that being up to one year older may entail more time of structured training (e.g., soccer academy), but is not necessarily linked to more unstructured sport experiences (e.g., street play, school play, family play). In this regard, our results are in line with the only direct attempt to relate the RAE to perceptual-cognitive performance in soccer players [65]. Seventy-six athletes from an elite soccer academy in Brazil (13 years old) completed a choice reaction time task, which showed no significant relationships between birth asymmetries, processing speed, and inhibitory control. The authors justified their findings by acknowledging that they were only able to evaluate soccer players from a state elite team. Hence, the pre-selection process of the soccer players may have been highly influenced by their maturation state, as coaches tend to select athletes at a more advanced maturational state [66]. Although our sample was composed of players from two different teams, both were elite teams that have more possibilities of selecting players than other teams from the region.

The quality of practice is another important factor that is related to the point raised previously. As some authors have highlighted, athletes in elite teams benefit from better training and competitive experiences and they have similar daily opportunities to fully develop themselves physically, technically, tactically and perceptual-cognitively [65]. Thus, in line with the literature relating soccer expertise (amateur vs. elite) to perceptual-cognitive benefits [62,63,64], our sample may show a group ceiling effect that is derived from high-quality practice. Regardless of their interindividual starting differences, elite athletes tend to reach similar cognitive benefits. Another possibility is that relatively younger players, in a context where they are not able to solve game situations by taking advantage of superior physical capacities, are forced to enhance their game intelligence, and subsequently their perceptual-cognitive capacities more. However, in our sample, coaches did not rate players that are born in the second half of the year as more or less intelligent, a result that is consistent with the reported lack of relationship between birth asymmetries and tactical performance in young players [67].

As other researchers have acknowledged [67], we were forced to simplify the study design due to the lack of time for the teams to participate in this type of evaluation in order to test elite Spanish soccer players. With this type of approaches, researchers are increasing the awareness of coaches, scouts, sport scientists, and academy directors regarding the strong influence of the RAE in sport selection and development. If we keep accomplishing this goal, step by step we will be able to control for variables whose evaluation is more time-consuming, such as the maturation stage of the players, a variable that has been strongly related to academy selection [68]. To properly answer the questions raised, there is a need for more longitudinal studies tracking soccer players born in different BQs from the beginning of their soccer experiences, also controlling for their unstructured sport experiences and other activities apart from sports in which they might have improved their perceptual-cognitive performance.

As happens with the vast majority of the literature on this topic, the cross-sectional design of our study does not allow us to test the nature vs. nurture issue. It is not clear whether the few players that were born in the last BQ were selected, because they already had excellent perceptual-cognitive abilities (to equate those of the players born in the first BQ) or whether their abilities had particularly improved (to equate those of players born in the first BQ), thanks to their interactions with older teammates and to their continuous efforts to adapt to a sport environment with higher competitive demands. It is likely that a complex combination of both factors combined with other modulating variables, emerged in the enriched environment of their sport.

It is important to take into consideration the results of our correlation analyses to further explore the relationship between the RAE, anthropometry, and physical and cognitive performance. Our findings suggest that this relationship is complex and multifactorial by nature. The age of athletes influences their anthropometric profile, which, as previously reported, impacts their physical performance. Superior physical capacities in athletes, such as cardiovascular fitness (i.e., endurance) or agility, has been also extensively related to cognitive performance. In fact, while our analysis did not show a strong relationship between the physical fitness variables that were assessed and most of the ANTI variables, in line with previous studies [31,59,60,61], it showed a link between all of the physical fitness variables evaluated and vigilance.

Future research is needed to better understand the various implications of RAE phenomena in youth sport development and their implications for cognitive performance. Prospective studies should include other general and sport-specific variables, together with the perception of coaches (e.g., scouts, academy directors) regarding potential for success. Skeletal age and individual maturation should also be considered to discern the relative contribution of chronological and biological age to youth sports development.

## 5. Conclusions

Although the age group differences (u10–u12) that were observed in all of the anthropometric, physical fitness and sustained attention variables were restricted to sample size and characteristics (i.e., elite club academies) of our study, they were not modulated by BQ. Similarly, the correlation between chronological age and anthropometrics, physical fitness, and sustained attention variables that were observed in the overall sample completely disappeared within each age group. Similarly, no BQ effects were found on the functioning of the attentional networks. Bayesian analyses supported the absence of BQ effects, corroborating that relative age was an inconsistent predictor of performance in the physical and perceptual-cognitive variables that were evaluated in the present study. 

Another crucial variable that may modulate the pattern of interrelations between the main variables of the present study is maturation. The present findings suggest that the selection process of soccer players may mitigate the age-related (i.e., BQ) differences that could be expected in some variables if the scouting process that lead to the birth asymmetries were not driven by short-term performance. The individual maturational state of players is likely to be one of the main variables that modulate this effect.

Our results, together with previous research, point to the importance of implementing various strategies to ensure equal formative opportunities in the youth sport system, such as *bio-banded* tournaments or teams grouped by athletes’ biological age [51,69]. In British soccer, for instance, as part of the Elite Player Performance Plan, a strategy for educating academy staff on the understanding of the growth and maturational processes has been promoted with the aim to better assess, monitor, and interpret differences in player development [70]; an example is the promotion of initiatives, such as “fourth quarter” trialist days, which were restricted to players born in the last BQ of the selection year [71]. Sport clubs prioritizing long-term players’ development instead of the desire for immediate competitive success will take physical biases in the selection and development of athletes into greater consideration.

## Figures and Tables

**Figure 1 ijerph-16-02837-f001:**
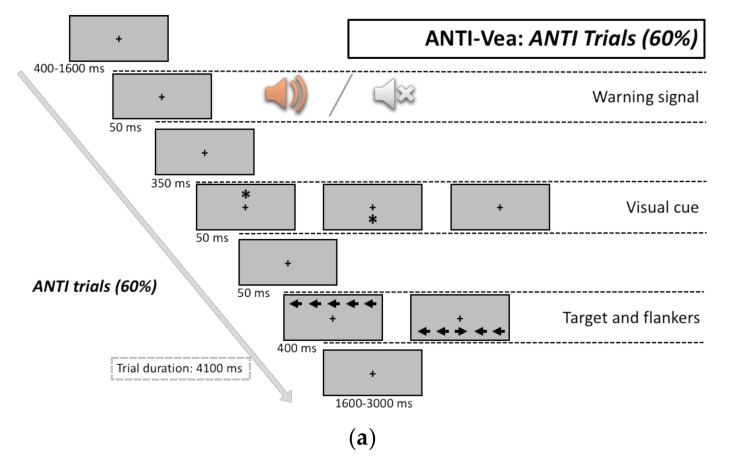
Experimental procedure and stimuli sequence in the Attentional Networks Test for Interactions and Vigilance—executive and arousal components (ANTI-vea) task: ANTI trials (**a**), executive vigilance (EV) trials (**b**), and arousal vigilance (AV) trials (**c**).

**Figure 2 ijerph-16-02837-f002:**
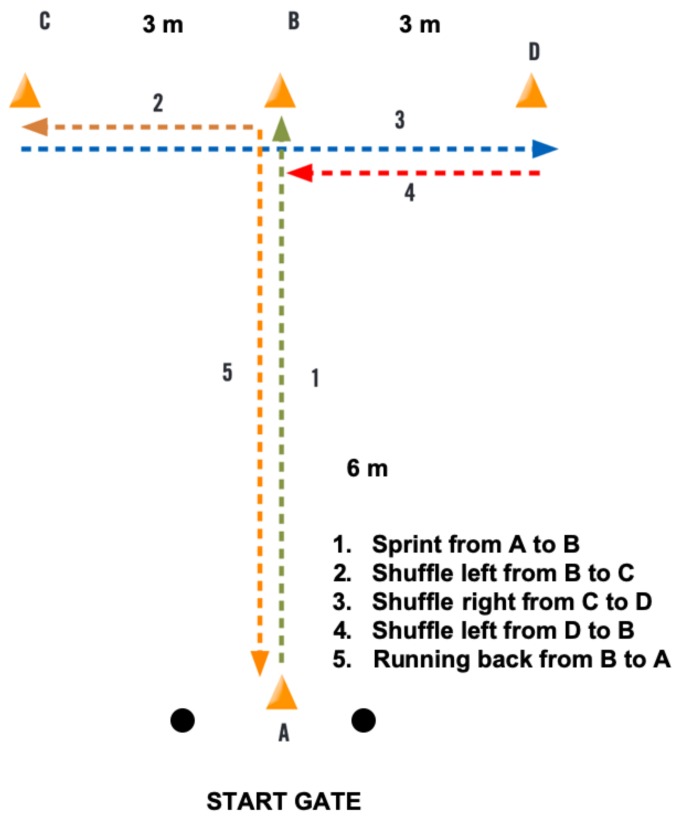
Set up of the modified T-test for agility measurement.

**Figure 3 ijerph-16-02837-f003:**
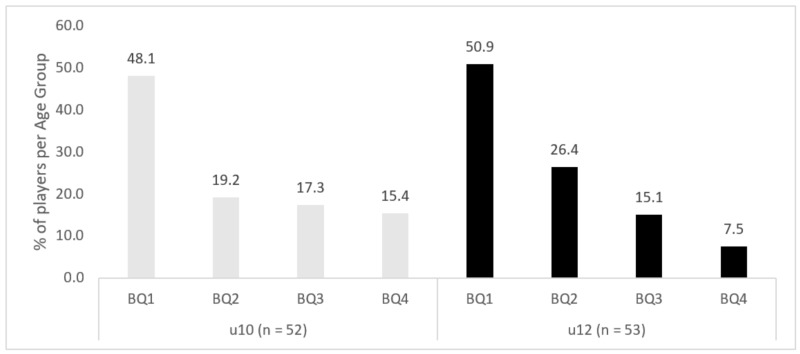
Percentage of players per Birth Quarter (BQ) and Age Group.

**Figure 4 ijerph-16-02837-f004:**
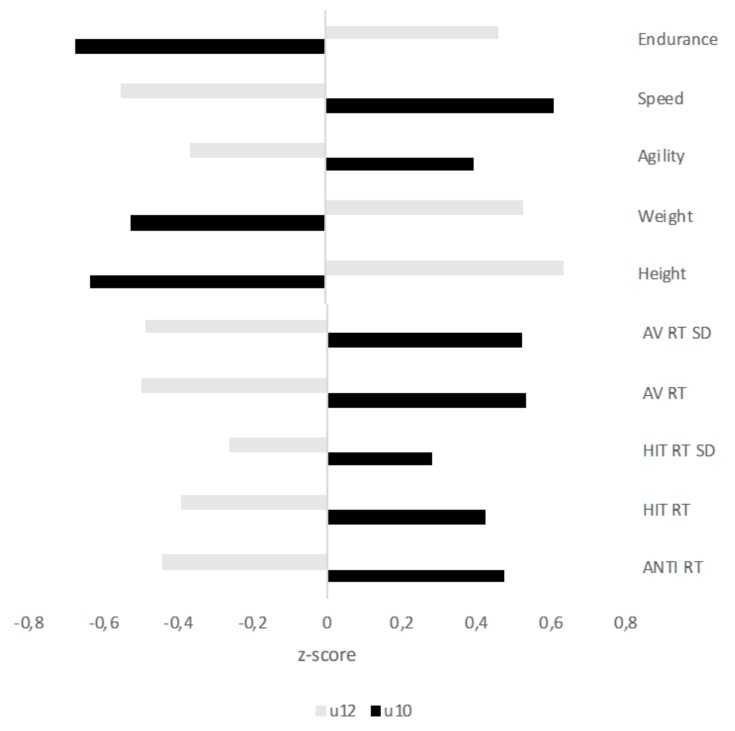
Mean standardized Z-scores for anthropometrics, physical fitness and cognitive measures showing a statistically significant main effect of Age Group.

**Table 1 ijerph-16-02837-t001:** Players’ characteristics (mean ± standard deviations) according to Age Group (u10 and u12) and Birth Quarter (BQ1, BQ2, BQ3, and BQ4).

			u10 (*n* = 52)				u12 (*n* = 53)				Age Group Effect		BQ effect	
Variables			BQ1 (*n* = 25)	BQ2 (*n* = 10)	BQ3v (*n* = 9)	BQ4 (*n* = 8)	BQ1 (*n* = 27)	BQ2 (*n* = 14)	BQ3 (*n* = 8)	BQ4 (*n* = 4)	F (*η_p_^2^*)	BF_10_	F (*η_p_^2^*)	BF_10_
Age & Anthropometrics		Chronological Age (y)	10.06 ± 0.09	9.83 ± 0.10	9.52 ± 0.13	9.35 ± 0.92	12.04 ± 0.08	11.8 ± 0.06	11.57 ± 0.07	11.27 ± 0.05	8667.39 (0.98) ***	>100	264.20 (0.89) ***	9.132
		Height (cm)	141 ± 6	142 ± 7	138 ± 4	135 ± 7	152 ± 9	151 ± 6	156 ± 12	148 ± 3	56.705 (0.381) ***	>100	1.694 (0.052)	0.65
		Weight (kg)	34.17 ± 4.79	35.96 ± 6.58	33.44 ± 4.14	32.80 ± 5.26	38.48 ± 6.81	40.03 ± 5.31	45.91 ± 11.32	39.65 ± 6.32	29.019 (0.240) ***	>100	0.661 (0.021)	0.133
Physical Fitness		Agility T-Test (sec)	8.09 ± 0.57	7.96 ± 0.38	8.00 ± 0.49	8.30 ± 0.68	6.87 ± 2.00	7.25 ± 0.42	7.56 ± 0.33	7.65 ± 0.29	6.699 (0.071) *	>100	0.598 (0.02)	0.157
		Speed 24 m (sec)	4.59 ± 0.24	4.57 ± 0.26	4.52 ± 0.18	4.55 ± 0.19	4.22 ± 0.27	4.19 ± 0.24	4.15 ± 0.26	4.40 ± 0.08	26.435 (0.233) ***	>100	0.666 (0.022)	0.114
		Endurance—TTE (min)	6.50 ± 2.37	6.40 ± 1.83	6.11 ± 1.75	6.76 ± 1.51	8.78 ± 1.55	9.62 ± 1.54	8.68 ± 1.41	8.16 ± 1.16	20.324 (0.220) ***	>100	0.313 (0.013)	0.187
Game Intelligence			3.19 ± 0.96	3.28 ± 1.08	3.24 ± 0.78	2.63 ± 1.01	3.25 ±0.93	3.20 ± 1.03	2.90 ± 1.45	2.65 ± 1.28	0.139 (0.001)	0.206	1.059 (0.032)	0.202
Attentional Functioning	ANTI trials	Mean RT ANTI (ms)	812 ± 176	7781 ± 142	864 ± 77	831 ± 125	723 ± 124	722 ± 11	659 ± 98	668 ± 58	13.937 (0.150) ***	83.896	0.088 (0.003)	0.086
		Mean RT SD ANTI (ms)	153 ± 38	143 ± 42	173 ± 12	162 ± 28	148 ± 34	149 ± 20	139 ±22	135 ± 10	3.341 (0.041)	0.5	0.243 (0.009)	0.097
		Mean Error ANTI Rate (%)	9.0 ± 9.4	14.6 ± 13.5	5.7 ± 2.4	5.2± 4.8	6.5 ± 4.5	6.6 ± 6.7	8.9 ± 8.5	7.8 ± 5.4	0.368 (0.005)	0.452	0.876 (0.032)	0.146
		Alertness Index RT (ms)	35 ± 53	56 ± 52	76 ± 61	51 ± 78	38 ± 68	48 ±65	63 ± 27	60 ± 42	0.024 (0.000)	0.226	1.143 (0.042)	0.282
		Alertness Index Error Rate (%)	−3.3 ± 11.7	5.4 ± 7.4	−0.2 ± 7.6	2.9 ± 2.7	1.3 ± 5.3	−0.5 ± 6.3	4.4 ± 6.3	−3.1 ± 12.4	0.109 (0.001)	0.27	1.007 (0.037)	0.185
		Orienting Index RT (ms)	23 ± 28	24 ± 63	33 ± 38	25 ± 41	34 ± 33	31 ± 31	44 ± 21	38 ± 27	1.288 (0.016)	0.501	0.286 (0.011)	0.102
		Orienting Index Error Rate (%)	1.9 ± 6.9	2.5 ± 4.6	−0.6 ± 2.8	2.3 ± 2.1	1.7 ± 6.5	0.9 ± 4.1	3.5 ± 3.4	5.3 ± 8.7	0.789 (0.016)	0.232	0.373 (0.014)	0.102
		Control Index RT(ms)	74 ± 49	80 ± 39	72 ± 23	79 ± 46	67 ± 42	69 ± 31	71 ± 18	51 ± 52	1.248 (0.016)	0.372	0.107 (0.004)	0.082
		Control Index Error Rate (%)	7.8 ± 8.6	15.2 ± 16.6	1.9 ± 5.2	4.5 ± 5.8	3.4 ± 6.3	5.9 ± 7.9	6.9 ± 6.5	3.9 ± 4.7	1.157 (0.014)	0.921	2.101 (0.174)	0.377
	VE Trials	Mean RT HIT (ms)	888 ± 2204	812 ± 203	999 ± 90	879 ± 145	799 ± 143	757 ± 118	730 ± 168	734 ± 116	11.119 (0.123) ***	20.971	0.763 (0.027)	0.172
		Mean RT SD HIT (ms)	173 ± 31	149 ± 55	181 ± 29	171 ± 29	160 ± 31	136 ± 24	155 ± 30	148 ± 18	4.852 (0.058) *	2.305	2.821 (0.093) *	1.589
		Hits (%)	64.9 ± 31.0	60.3 ± 36.5	80.0 ± 18.2	78.9 ± 19.4	70.9 ± 27.5	74.6 ± 21.3	64.8 ± 26.2	65.1 ± 24.8	0.092 (0.001)	0.24	0.159 (0.006)	0.092
		False Alarms (%)	7.1 ± 10.1	8.4 ± 10.7	5.8 ± 5.5	6.0 ± 4.6	5.7 ± 4.4	3.4 ± 4.6	6.6 ± 6.6	7.6 ± 6.4	0.307 (0.004)	0.366	0.079 (0.003)	0.085
	AV trials	AV RT (ms)	661 ± 105	610 ± 78	653 ± 118	602 ± 73	556 ± 93	536 ± 82	542 ± 54	547 ± 116	22.22 (0.207) ***	>100	0.697 (0.025)	0.172
		AV SD (ms)	127 ± 54	98 ± 24	136 ± 43	133 ± 43	98 ± 33	90 ± 35	72 ± 16	66 ± 9	15.37 (0.153) ***	>100	0.844 (0.03)	0.204

Note: Results of ANOVAs (*F* value and estimated effect sizes) and Bayesian analyses (Bayes factors, BF_10_) are included. * *p* < 0.05, ** *p* < 0.01, *** *p* < 0.001.

## Data Availability

The datasets generated during and/or analysed during the current study are available in the OSFHOME repository, DOI:10.17605/OSF.IO/5RPCQ https://osf.io/5rpcq/?view_only=eee4e4f021fc4583b64d09898b2e66f4.

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
