# Peer review of "Relative Age Effect in the Sport Environment. Role of Physical Fitness and Cognitive Function in Youth Soccer Players"

_ijerph, 2019, doi:10.3390/ijerph16162837_

Round 1
Reviewer 1 Report
This article examines Relative Age Effects among elite youth soccer/football players with samples of U10 (under 10) and U12 (under 12) programs and integrates variables that are not typical in prior work that usually relies exclusively on publicly available observational data (among older players). The writing of the article needs substantial improvement throughout the paper; however, strengths of the content include its review of prior work as well as important aspects of the new study (e.g., the authors find substantial selection in favor of first-quartile players and correspondingly no substantial relation between RAE and the various measures of physical and cognitive performance they report).
More specific aspects to highlight for the authors include:
+ False Alarms needs to be defined by the time of its first appearance in the article. Indeed, the authors would be well-served by reviewing all such terms to ensure that definitions are presented upon first mention of given variables/measures.
+ Additional papers on RAE that would carry relevance for the work include both the reviews and findings presented in these works (listed alphabetically):
Fumarco, L. (2015). Essays on discrimination in the marketplace (Doctoral dissertation, Linnaeus University Press).
Heneghan, J. F., & Herron, M. C. Relative age effects in American professional football. Journal of Quantitative Analysis in Sports.
Kniffin, K. M., & Hanks, A. S. (2016). Revisiting Gladwell's hockey players: Influence of relative age effects upon earning the PhD. Contemporary Economic Policy, 34(1), 21-36.
+ Do the authors consider the high percentage of 1Q players to represent a mistake or inefficiency in the selection system of the teams/programs? Perhaps especially in relation to some of the prior work on RAE, this kind of question seems to warrant greater attention in light of the authors' work.
+ Can the authors acknowledge the parents (and coaches) whose decisions have relevance to the players or study participants? Minimally, identification of the parents in relation to the study (and any informed consent that was provided) would be helpful. Further, though, the roles of parents (and coaches) seems important to acknowledge given the young ages of the players who are studied in the current work.
Author Response
Dear Ms. Sabrina Sun/MDPI Assistant Editor,
Dear Reviewer 1,
Please find enclosed a revised version of our manuscript entitled “Relative age effect in sport environment. Role of physical fitness and cognitive function in youth football players” which we originally submitted for publication in International Journal of Environmental Research and Public Health (Manuscript ID: IJERPH-534551)
We are pleased that the reviewer and member of Editorial Board found our work of interest. Moreover, we thank both Reviewers for the suggestions to improve the quality of our previous submission. We appreciate the opportunity to revise and resubmit our manuscript.
Below, we have provided a point-by-point response (Author response: AR) to the issues raised by the reviewer (R). We have also highlighted relevant sections of the manuscript to facilitate consideration of the revisions.
We feel the manuscript has improved as a result of this initial review and hope the paper is now suitable for publication in the International Journal of Environmental Research and Public Health.
REVIEWER 1
Comments and Suggestions for Authors
Point 1: The writing of the article needs substantial improvement throughout the paper; however, strengths of the content include its review of prior work as well as important aspects of the new study (e.g., the authors find substantial selection in favor of first-quartile players and correspondingly no substantial relation between RAE and the various measures of physical and cognitive performance they report).
Response 1: We really appreciate your considerations about the strengths of the first draft submitted and we will try to improve the manuscript using your meaningful contributions.
Point 2: False Alarms needs to be defined by the time of its first appearance in the article. Indeed, the authors would be well-served by reviewing all such terms to ensure that definitions are presented upon first mention of given variables/measures.
Response 2: In the first draft submitted for review, the term “False Alarms” was used in the first time at the page 4, line 188. Definition of this term was described on the next lines 185-187 as follows: “False Alarms (FAs) were defined as the proportion of space bar responses (i.e., the response for infrequent stimuli) given to non-displaced targets”. We have now checked that all variables and measures have been accurately defined at their first mention in the manuscript.
Point 3: Additional papers on RAE that would carry relevance for the work include both the reviews and findings presented in these works (listed alphabetically):
· Fumarco, L. (2015). Essays on discrimination in the marketplace (Doctoral dissertation, Linnaeus University Press).
· Heneghan, J. F., & Herron, M. C. Relative age effects in American professional football. Journal of Quantitative Analysis in Sports.
· Kniffin, K. M., & Hanks, A. S. (2016). Revisiting Gladwell's hockey players: Influence of relative age effects upon earning the PhD. Contemporary Economic Policy, 34(1), 21-36.
Response 3: We have reviewed the literature related with RAE, and specifically those related with its implications with physical and cognitive capabilities. Given that RAE has been studied in a vast of different contexts, and in order to focus our research interest, we had to prioritize and filter the literature more directly related with our aims and goals. Nevertheless, taking into account the reviewer´s suggestion we appreciate some of his/her suggestions and we have now completed the reviewed literature including some new updated references related with the suggested previously by the reviewer:
- Fumarco, L., & Rossi, G. (2018). The relative age effect on labour market outcomes – evidence from Italian football. European Sport Management Quarterly, 18(4), 501-516. https://doi.org/10.1080/16184742.2018.1424225 (see page 2, lines 70-71)
- Heneghan J. F., & Herron M. C. (2019). Relative age effects in American professional football. Journal of Quantitative Analysis in Sports, 0(0). https://doi.org/10.1515/jqas-2018-0030. (see page 2, line 66)
However, following also the comment 3 by reviewer 2, we decided not to include the reference by Kniffin and Hanks (2016) because it is not directly related with sport context, but academic (the influence of RAE on salary of PhD).
Point 4: Do the authors consider the high percentage of 1Q players to represent a mistake or inefficiency in the selection system of the teams/programs? Perhaps especially in relation to some of the prior work on RAE, this kind of question seems to warrant greater attention in light of the authors' work.
Response 4: We appreciate the interesting reflection by the reviewer. As we have shown in the last part of the introduction, we hypothesized that RAE (asymmetrical distribution of players based on the month of birth) should be reflected on a maturity bias (i.e. greater size, strength, speed and power) and therefore, we could expect better physical and cognitive performance in those players born in the first half of the year.
This hypothesis could be supported by the fact that we consider that people responsible of the selection–scouting process in elite football academies are not giving relevance to the month of birth or maturational status of the players, but their actual talent in different variables associated to sport-specific performance. Our findings have shown no differences in performance in none of the selected variables, and therefore, as we have reflected in the discussion and conclusion sections, the selection process mitigates the age-related (BQ) differences that could be expected in some variables if the scouting process leading to the birth asymmetries were not driven by short-term performance.
Then, our response to the reviewer´s question is that the high percentage of 1Q players at these ages could be a mistake produced by an inefficient procedure if the goals were to select players looking for medium-long term goals, but it could be indeed considered a very successful procedure when immediate or short-term goals are pursued. According to previous literature, players’ individual maturational state may be one of the main variables modulating the RAE. Therefore, the mistake would be to look for very short-term performance, which inevitable lead to the RAE as a way to equate team players from different month of birth.
Following the reviewer´s suggestions we have included a paragraph highlighting this issue on the introduction section (see page 2 & 3, lines 94-97).
Point 5: Can the authors acknowledge the parents (and coaches) whose decisions have relevance to the players or study participants? Minimally, identification of the parents in relation to the study (and any informed consent that was provided) would be helpful. Further, though, the roles of parents (and coaches) seems important to acknowledge given the young ages of the players who are studied in the current work.
Response 5: We have now modified the “Acknowledgement” section including a specific mention to player´s parents and coaches (see page 15, lines 534-535).

Reviewer 2 Report
This is an interesting study with practical applications. Although the authors have used correct scientific methods, there are several issues that they need to address before publishing their paper. A major issue is the language that must be revised because it is difficult to follow some parts of the text. For more details, please revise according my specific comments below.
1. Revise all text for English.
2. RAE has been well studied recently by a large number of studies (https://www.ncbi.nlm.nih.gov/pubmed/?term=%22relative+age+effect%22+AND+soccer); thus, the focus of the paper should be shifted to the cognitive aspects.
3. Abstract: Improve the rationale/aim for this study.
4. Abstract: Present more numbers, mean, SD, p values, effect sizes for comparisons.
5. Introduction: Revise this part according to my comment No2.
6. Methods, l.110: change 0.5 to 0.05
7. Throughout the text, emerge small paragraphs of 2-3 lines into larger one.
8. Results, tables: Use . instead of , for decimals
9. What about the relationship of cognitive function with fitness and anthropometry? E.g. it has been shown that fitness is associated with anthropometry (see Nikolaidis PT, Elevated body mass index and body fat percentage are associated with decreased physical fitness in soccer players aged 12-14 years. Asian J Sports Med. 2012 Sep;3(3):168-74; 154. Nikolaidis, P.T. (2013): Prevalence of overweight and association between body mass index, body fat percent and physical fitness in male soccer players aged 14-16 years. Science & Sports, 28(3):125-132. doi: 10.1016/j.scispo.2012.12.002.)
Author Response
Dear Ms. Sabrina Sun/MDPI Assistant Editor,
Dear Reviewer 2,
Please find enclosed a revised version of our manuscript entitled “Relative age effect in sport environment. Role of physical fitness and cognitive function in youth football players” which we originally submitted for publication in International Journal of Environmental Research and Public Health (Manuscript ID: IJERPH-534551)
We are pleased that the reviewer and member of Editorial Board found our work of interest. Moreover, we thank both Reviewers for the suggestions to improve the quality of our previous submission. We appreciate the opportunity to revise and resubmit our manuscript.
Below, we have provided a point-by-point response (Author response: AR) to the issues raised by the reviewer (R). We have also highlighted relevant sections of the manuscript to facilitate consideration of the revisions.
We feel the manuscript has improved as a result of this initial review and hope the paper is now suitable for publication in the International Journal of Environmental Research and Public Health.
REVIEWER 2
Comments and Suggestions for Authors
Point 1: This is an interesting study with practical applications. Although the authors have used correct scientific methods, there are several issues that they need to address before publishing their paper. A major issue is the language that must be revised because it is difficult to follow some parts of the text. For more details, please revise according my specific comments below.
Response 1: We really appreciate your considerations about the methodology and interest and practical applications of our research. According to your suggestions about language and other above-mentioned weakness issues, we have now tried to improve our first draft submitted using your meaningful contributions.
Point 2: Revise all text for English.
Response 2: The new version of the manuscript has been rigorously revised to improve the English.
Point 3: RAE has been well studied recently by a large number of studies (https://www.ncbi.nlm.nih.gov/pubmed/?term=%22relative+age+effect%22+AND+soccer); thus, the focus of the paper should be shifted to the cognitive aspects.
Response 3: We agree with the reviewer´s opinion about the extent of the study of RAE in football. However, and as we highlighted throughout the introduction section, most of the studies focused on the description of the presence or the absence of this selection bias in different context and ages. As we have detailed in the last part of the introduction (aims & goals paragraphs) here we were interested into the knowledge about the behavioral (not only, but importantly cognitive functioning) relevance of RAE on different variables related with sport performance (anthropometrics, physical fitness, game intelligence and attentional functioning).
Therefore, following the reviewer’s advice we have tried to highlight the contribution of our paper by incorporating the cognitive aspects, although not only them, especially when we used an experimental-task measuring different aspects of attention and vigilance, not a sport-specific attentional task. However, the importance of our work might be that we do so in combination with more soport-based measures, which increase the relevance and comprehension of the RAE phenomena at these early stages of football players´ selection and training process.
Point 4: Abstract: Improve the rationale/aim for this study.
Response 4: Thanks for the suggestion. Due to the word limit in the abstract it is difficult to explain in a deeper way our rationale. However, we have tried to explain further the aim of the study in the introduction (page 1, lines 15-17)
Point 5: Abstract: Present more numbers, mean, SD, p values, effect sizes for comparisons.
Response 5: We have included now some statistics (p- values and BF10; not all, in order to accomplish the recommendations about the 200 words abstract´s length) in order to improve the reader´s knowledge about the statistical relevance of our main findings.
Point 6: Introduction: Revise this part according to my comment No3.
Response 6: We thank the reviewer for the interesting point raised. We have completed the introduction with relevant literature exploring this issue (page 2, lines 70-71).
Point 7: Methods, l.110: change 0.5 to 0.05
Response 7:Done.
Point 8: Throughout the text, emerge small paragraphs of 2-3 lines into larger one.
Response 8:Your recommendation has been implemented throughout the manuscript.
Point 9: Results, tables: Use . instead of , for decimals
Response 9:Done.
Point 10: What about the relationship of cognitive function with fitness and anthropometry? E.g. it has been shown that fitness is associated with anthropometry (see Nikolaidis PT, Elevated body mass index and body fat percentage are associated with decreased physical fitness in soccer players aged 12-14 years. Asian J Sports Med. 2012 Sep;3(3):168-74; 154. Nikolaidis, P.T. (2013): Prevalence of overweight and association between body mass index, body fat percent and physical fitness in male soccer players aged 14-16 years. Science & Sports, 28(3):125-132. doi: 10.1016/j.scispo.2012.12.002.)
Response 10:.
We totally see the point raised by the reviewer. It is very interesting indeed. However, in the study we present multiple results linked to our primary objectives, that is why we decided not to extent more the results section with results that are not of primary relevance for the study. Future research should also incorporate other fitness measures and direct evaluation of individual´s maturation state.

Round 2
Reviewer 2 Report
Unfortunately, the language has not been improved to increase the readability of the text.
In addition, the authors did not address successfully some points.
I recommend rejection and I encourage the authors to re-submit after having considered seriously my comments.
Author Response
Dear Ms. Sabrina Sun/MDPI Assistant Editor,
Dear Academic Editor and Reviewer 2,
Please find enclosed a re-revised version of our manuscript entitled “Relative age effect in the sport environment. Role of physical fitness and cognitive function in youth football players”, which we originally submitted for publication in the International Journal of Environmental Research and Public Health (Manuscript ID: IJERPH-534551) on 10 June 2019, and resubmitted with changes on 28 June 2019.
We are pleased that Reviewer 1 was satisfied with our answers. However, although we welcome the fact that Reviewer 2 recognized the scientific value of our study, we regret not having been able to satisfy his/her suggestions on some relevant points. In this regard, we appreciate the opportunity granted to us by the Academic Editor and reviewer to revise and resubmit our manuscript again.
Below we provide a point-by-point response to the issues raised by the reviewer (R) and Academic Editor (AE). We have also highlighted changes in relevant sections of the manuscript to facilitate further consideration of the revisions.
We feel the manuscript has improved as a result of both previous reviews and hope the paper is now appropriate for publication in the International Journal of Environmental Research and Public Health.
REVIEWER 2 & ACADEMIC EDITOR
Comments and Suggestions for Authors
Point 1: R2: "Revise text for English".
AE: It is not my mother tongue, but I recognize that the writing can be improved. Please ask a colleague, who is mother tongue (UK, USA) to provide language editing.
Response 1: We apologize to you for all the language mistakes. The present resubmitted manuscript has been revised by a professional translator who is a native English speaker with broad experience in translations in the area of psychology. We have not marked up in the text all the editing changes related with the use of English language in order to make the reading easier for the editors and reviewers.
Point 2: R2: RAE has been well studied recently by a large number of studies; thus, the focus of the paper should be shifted to the cognitive aspects. (...) Introduction: Revise this part according to my comment No3." Comment No3.: RAE has been well studied recently by a large number of studies. (https://www.ncbi.nlm.nih.gov/pubmed/?term=%22relative+age+effect%22+AND+soccer); thus, the focus of the paper should be shifted to the cognitive aspects.
AE: The information you provided in lines 70-71 is too poor. You have seemingly not addressed this point, which could emphasize the originality and novelty of your study, that is the added value of you work for the RAE literature. You may mention general literature on fitness and cognition/attention to move forward to last reviews on the major role of sport/expertise than physical fitness in determining an advantage in both domain-general and sport-specific cognition and attention in open-skill athletes. So far as regards the introduction. This point can be easily linked to the other point you have seemingly neglected.
Response 2: According to both comments by the editor and reviewer and in order to define our framework more clearly (relative age effect on cognition and more specifically on attentional functioning), we have now changed some paragraphs in different sections of the manuscript.
· We have now highlighted more clearly in the abstract and in the last paragraph of the introduction, including the aims and goals (see page 3, lines 112 to 122) the originality and novelty of our study, focusing on the study of the RAE on different behavior´s components, and especially on cognitive-attentional functioning.
· We have re- structured the introduction moving, removing and/ or shortening some paragraphs related to general literature about the RAE (maintaining some information about this issue that were required by Reviewer 1 during 1st round of review).
· We have now cited some general and specific studies analyzing the role of physical fitness and sport expertise on cognitive/attentional functioning, showing the more significant role of the type of sport practice (open-skill sports) compared to physical fitness in developing both domain-general and sport-specific cognitive skills (See pages 2, lines 88-94).
Point 3: R2: What about the relationship of cognitive function with fitness and anthropometry? E.g. it has been shown that fitness is associated with anthropometry (see Nikolaidis PT, Elevated body mass index and body fat percentage are associated with decreased physical fitness in soccer players aged 12-14 years. Asian J Sports Med. 2012 Sep;3(3):168-74; 154. Nikolaidis, P.T. (2013): Prevalence of overweight and association between body mass index, body fat percent and physical fitness in male soccer players aged 14-16 years. Science & Sports, 28(3):125-132. doi: 10.1016/j.scispo.2012.12.002.)
AE: Following the reviewer’s suggestion, you may emphasize the value of your work. In the introduction, you properly mention that whereas the anthropometric maturation seems to lead the early selection process, there are other relevant physical and cognitive factors relevant to performance development which, being neglected in favor of early maturation signs, lead to suboptimal selection at the expense of potential talents, who were born in the last months of the year. The Reviewer asks you now to consider not only those relevant physical and cognitive factors individually, but also their interconnectedness. Exercise and cognition research provides evidence of this complex pattern of interrelations, which should be considered in early selection rather than mere anthropometric measures of early maturation. Integrating this view in the Introduction, Results and Discussion sections in response to the request of the reviewer (hoping that I have interpreted it correctly), can enhance the informative value of your work and lead to the publication of a stronger paper.
Response 3: We totally understand the point raised by the reviewer and academic editor. It is very interesting indeed. We have tried to include all the suggestions by the reviewer and academic editor in the next manner:
· We have now included in the introduction section (see page 2, lines 83-94) few sentences about the interrelations between anthropometrics (body mass), fitness and cognition that could influence the athletic success. Here we have highlighted that
· Regarding the specific suggestion of the reviewer about the relationship between cognitive function with fitness and anthropometry, we have now highlighted the importance of studying their interconnections in order to improve the knowledge about the complexity of the scouting of young talents in soccer. In this vein we appreciate the recommendation by the reviewer to justify this relationship between anthropometrics and physical fitness (see cites 27 and 28 in page 2, lines 93-94).
· Here we have highlighted that this complex pattern of interrelations between variables should be considered in early scouting rather than mere anthropometric or physical fitness indicators of early maturation (page 3, lines 106-111).
· In the Results section we presented multiple results linked to our primary objectives. This is why we decided not to extend the Results section more with results that were not initially considered of primary interest in this research due to our study design. However, we totally understand the value of seeking a broader picture of the sport-cognition relationship. Hence, following your recommendations we have added further analyses (see the Table 2 in Supplementary Material) to better explore the interconnections between cognitive function, fitness and anthropometrics.
· Previous results about interconnections between cognitive function, fitness and anthropometrics have been discussed in the Discussion section (see page 15, lines 520-528), and including few sentences in the Conclusion section (see page 15, lines 539-541).

Round 3
Reviewer 2 Report
The authors addressed successfully all my concerns. Thus, I recommend this work to be published.